# Autologous micro-fragmented adipose tissue in the treatment of atherosclerosis patients with knee osteoarthritis in geriatric population: A systematic review and meta-analysis

**Wei Li[1], Huajuan Guo[1], Congcong Wang[1], Yimin Zhang[1], Jun Wang[1,2,3]***

**1** Department of Joint Surgery, Weifang People's Hospital, Weifang, China, **2** School of Rehabilitation Medicine, Capital Medical University, Beijing, China, **3** China Rehabilitation Research Center, Beijing Bo'ai Hospital, Beijing, China

* abcd2878@126.com

## Abstract

### Background

Atherosclerosis and osteoarthritis are closely related. However, no high-quality studies have investigated the potential of micro-fragmented adipose tissue to treat patients with atherosclerosis accompanied by osteoarthritis.

### Methods

PubMed, Embase, the Cochrane Library, Web of Science, China National Knowledge Infrastructure, WANFANG DATA, and CQVIP were searched for potentially eligible studies published before October 13, 2022. Due to the statistical limitations of the existing relevant literature, it is not possible to make direct statistics on the patients with osteoarthritis accompanied by atherosclerosis treated by micro-fragmented adipose tissue. The primary outcome consisted of two parts: 1) Correlation between atherosclerosis and osteoarthritis; 2) Scores of the Knee injury and Osteoarthritis Outcome Score (KOOS). And secondary outcomes were pain assessed by visual analog scale (VAS) or numeric rating scale (NRS), quality of life (QoL) (assessed using tools apart from the KOOS), and adverse events (AEs). Random meta-analysis was conducted using STATA 14.0.

### Results

Nineteen studies were included. The metaanalysis evidenced a positive association between atherosclerosis and osteoarthritis (OR 1.17, CI 1.01–1.36). The mean absolute difference in KOOS subscale scores between pre- and post-treatment (mean with 95% confidence interval [CI]) was 19.65 (13.66, 25.63), 14.20 (4.64, 23.76), 19.95 (13.02, 26.89), 25.23 (14.80, 35.66), and 26.01 (13.68, 38.35) for pain, symptoms, activities of daily living (ADL), sports/recreation, and quality-of-life (QOL), respectively. The mean differences in

**Data Availability Statement:** All relevant data are within the paper and its Supporting Information files.

**Funding:** This study was supported in the form of funding by Shandong Province Natural Science Foundation (Grant No. ZR2019PH026 and ZR2023MH145) awarded to JW, China University Industry-University-Research Innovation Fund (Grant No. 2023HT050) awarded to JW, Shandong Province Natural Science Foundation (Grant No. ZR2019PH026) awarded to YZ, and Weifang City Science and Technology Development Plan Project Fund (Grant No. 2020YX014) awarded to WL.

**Competing interests:** The authors have declared that no competing interests exist.

VAS, resting VAS, activity VAS, and NRS between pre- and post-treatment was -8.24 (-10.66, —5.82), -3.61 (-4.49, -2.72), -4.17 (-4.89, -3.44), and -2.17 (-4.06, and -0.28), respectively. The mean difference in score of Western Ontario and McMaster Universities Osteoarthritis Index (WOMAC), EQ-5D, and University of California in Los Angeles (UCLA) between pre- and post-treatment was -24.81 (-40.80, -8.82), 0.07 (0.02, 0.12), and 0.30 (-0.42, 1.02), respectively. The mean difference in Tegner score and the International Knee Documentation Committee (IKDC) score between pre- and post-treatment was 0.67 (-0.62, 1.97) and 13.70 (6.35, 21.04), respectively. The use of micro-fragmented adipose tissue was associated with risk of bruising, bleeding, hematoma, drainage, infection, soreness, swelling, pain, and stiffness in harvest and injection sites.

## Conclusion

Atherosclerosis and osteoarthritis share common risk factors and comorbidity. And the use of micro-fragmented adipose tissue may benefit for improving symptoms of knee osteoarthritis accompanied by atherosclerosis although may lead to some mild adverse events. Randomized controlled trials with long-term follow-up are necessary for further evaluation because many limitations of this meta-analysis cannot be ignored.

## 1. Introduction

Knee osteoarthritis (OA) is one of the most prevalent musculoskeletal disorders, with a significant impact on one's ability to carry out daily activities (ADL) [1]. Knee OA is characterized by articular cartilage loss, bone remodeling, and periarticular muscle weakness, resulting in knee joint pain, swelling, deformity, and instability [2, 3]. Over half of the persons suffering from symptomatic knee OA experience significant disabilities daily [4]. It cannot be denied that knee OA has risen to become the leading cause of disability among the elderly [5]. In United States of America Studies, 38% to 47% of elderly aged more than 60 years being affected by knee OA [6, 7]. In addition, according to recent studies, 18% of elderly males and 27% of elderly females were estimated to have KOA, and this ratio is estimated to increase over the coming decades [8]. Therefore, it is crucial to early use effective therapies to treat knee OA.

Cardiovascular disease (CVD) remains the leading cause of death worldwide, and atherosclerosis is the primary pathophysiological substrate of the disease process. Osteoarthritis (OA) is the most common cause of joint disease worldwide and its incidence is increasing as the population ages, affecting one in five people worldwide. As two very common diseases, their frequent co-existence is not surprising and increases the likelihood that they will interact with each other. Not surprisingly, OA has been identified as a credible risk factor for non-classical CVD. The association between osteoarthritis and cardiovascular disease is complex, difficult to direct, and stems from a common mechanism in the origin of chronic low-grade inflammation, compounded risk factors, and lifestyle.The prevalence of these two diseases in the general population is strongly related to age. For osteoarthritis, the prevalence increases in people over the age of 40. However, the increase in prevalence was different between the sexes. After the age of menopause, prevalence increases twice as fast in women as in men. Similar sex-specific differences were seen in atherosclerotic related deaths. The leading cause of death in men after the age of 35 is associated with atherosclerosis, while the number of atherosclerotic deaths in women begins to increase mainly after the age of menopause and becomes the

leading cause of death after the age of 60. In old age, women have an even higher risk of death from atherosclerosis than men. In addition to the common risk factors for atherosclerosis and OA such as age and BMI, systematic reviews show that typical risk factors for atherosclerosis are most likely also risk factors for OA, such as high blood pressure for OA of the knee and high cholesterol levels for OA of the knee.

The primary goals of treatment for knee OA are to relieve pain and inflammation, reducing stiffness, optimizing mobility, function, and range of motion, and improving quality of life (QoL) [3, 9]. Currently, non-surgical and surgical therapies are available for the treatment of knee OA [10]. Non-surgical therapies include physical therapy, oral medicines, and intra-articular injections of steroids, hyaluronic acid, and platelet-rich plasma (PRP); nevertheless, these are only capable of providing limited therapeutic benefits, with an effect that is frequently not totally satisfactory and even decreasing over time, and varies between patients [11–13]. For more advanced diseases, joint-preserving and joint-replacing surgeries can be considered [3, 9]. As the ultimate therapeutic option for knee OA, primary and revision total knee arthroplasties (TKA) carry a non-negligible risk of failure and the need for later revision arthroplasty, especially in younger patients [14]. Therefore, it is imperative to find new effective treatment options for knee OA in order to delay or prevent invasive surgery [15].

Recently, the use of mesenchymal stromal cells (MSCs) has been introduced as an attractive therapeutic option for knee OA due to their immunomodulatory, anti-inflammatory, and paracrine properties [16, 17]. Bone marrow and adipose tissue were considered as the most common sources of MSCs [18], and adipose tissue-derived MSCs have similar properties to bone marrow-derived MSCs [19]; however, adipose-derived MSCs are easier to collect for clinical applications and have a higher isolation rate than bone marrow-derived MSCs [19–21]. Furthermore, micro-fragmented adipose tissue (MFAT) has been described as one of the smartest and most straightforward ways to employ adipose tissue in regenerative techniques in a variety of clinical applications [22, 23]. Specifically, MFAT contains MSCs that originate from the adipose blood vessels as pericytes and are released and primed during the extraction process through sheer stress and microfiltration of the adipose tissue [24, 25]. MFAT would be beneficial in providing an optimal biological environment for healing because its advantages in providing a high number of cells and growth factors without expansion or enzymatic treatment, thus preserving the integrity of cells and tissue microarchitecture [26]. To date, several studies have shown promising results when using autologous MFAT injection to treat knee OA [27, 28]. In animal experiments, it was found that MFAT produced by Lipogems achieved the best cartilage repair effect in the treatment of knee osteoarthritis with enzymatic digestion or cell expansion [29]. Zeira showed that MFAT injection in 130 dogs with spontaneous osteoarthritis is safe, feasible, and beneficial [30]. In clinical treatments, Russo evaluated the 1-year safety and outcome of a single intra-articular injection of autologous microfragment adipose tissue in 30 patients with knee osteoarthritis. An overall median improvement of 20 points was observed in IKDC subjective and total KOOS, and higher success rates were found in VAS pain and Tegner Lysholm knee joints, of which 22 patients maintained good outcomes at follow-up 3 years later with no associated complications or clinical deterioration documented [31, 32]. And another study showed significant improvements in pain, quality of life, and function for at least 12 months in patients with refractory severe (grade 3 or 4) knee osteoarthritis [33]. A multi-centric, international study show that seventy-five patients, 120 primary treatments, with KL grade 2 to 4 knee OA treated with a single MFAT injection. Patients with KL grade 2 disease had the best results in KOOS—Pain. Including advanced KL grade 3 and 4 osteoarthritis patients, significant functional and quality of life success was seen in 106/120 treatments at all follow-up time points. Fourteen treatments failed prior to the study endpoint [34]. In addition, a number of literature studies have found that MFAT has a good effect on

knee/hip osteochondral injury and knee/hip osteoarthritis [35, 36]. And the clinical efficacy is better than repeated doses of leucocyte-poor platelet-rich plasma plus hyaluronic acid [37].

Still, to date, no high-level studies have comprehensively investigated the therapeutic efficacy and safety of autologous MFAT in the treatment of knee OA. Therefore, we conducted the current meta-analysis to systematically evaluate the therapeutic efficacy and safety of autologous MFAT for the treatment of knee OA, with the aim to help increase the focus on this strategy for performing additional studies to improve patient prognosis.

## 2. Materials and methods

### 2.1. Literature search

This systematic review and meta-analysis was conducted according to the Preferred Reporting Items for Systematic Reviews and Meta-Analyses (PRISMA 2020) guidelines [38, 39]. The study was designed based on the PICOS principle [40]. Two investigators independently searched PubMed, Embase, the Cochrane Library, Web of Science, China National Knowledge Infrastructure (CNKI), WANFANG DATA, and CQVIP for potentially eligible studies published before October 13, 2022, using the medical subject heading (MeSH) terms of 'Osteoarthritis', 'Osteoarthritis, Knee', and 'Autologous Micro-fragmented Fat Tissue' as well as relevant analogs. The retrieved records were screened according to the eligibility criteria. The detailed search strategies of target databases are summarized in S1 Table. Any discrepancies between the two investigators were resolved by discussion until a consensus was reached. Due to the statistical limitations of the existing relevant literature, it is not possible to make statistics on the patients with OA accompanied by AT treated by MFAT. So we made the primary outcome consist of two parts: 1) Correlation between atherosclerosis and osteoarthritis; 2) Scores of the Knee injury and Osteoarthritis Outcome Score (KOOS).

### 2.2. Eligibility criteria

The inclusion criteria were as follow: (1) population: knee OA, (2) intervention: surgery combined with autologous MFAI, (3) comparison: not specifically defined, (4) outcomes: the Knee injury and Osteoarthritis Outcome Score (KOOS), pain, quality of life (QoL) (assessed using tools apart from the KOOS), and adverse events (AEs), and (5) full texts published in English and Chinese.

### 2.3. Exclusion criteria

The exclusion criteria were (1) in vitro study, (2) animal experiment, or (3) studies with ineligible designs, such as meta-analysis, review, case report, commentary, letter to the editor, or conference abstract.

### 2.4. Data extraction and synthesis

Data extraction was conducted by two investigators. The extracted data included study characteristics (authors, year of publication, country where the study was performed, and study design), patient characteristics (sex, sample size, and knee OA degree), intervention characteristics (procedure, donor site for adipose tissue, and device), and outcome data. The primary outcome was the KOOS, which includes five subscales: pain, symptoms, activities of daily living (ADL), sports/recreation, and QoL. The continuous variables in the form of mean ± standard deviation (SD) at the last follow-up were extracted. The secondary outcomes were pain assessed by visual analog scale (VAS) or numeric rating scale (NRS), QoL (assessed

using tools apart from the KOOS), and AEs. Disagreements between the two investigators were resolved by discussion until reaching a consensus.

## 2.5. Quality of the evidence

The level of evidence of all studies was assessed independently by two investigators according to the Methodological Index for Non-Randomized Studies (MINORS) [41]. The evaluation is based on (1) a clearly stated aim, (2) the inclusion of consecutive patients, (3) a prospective collection of data, (4) appropriate endpoints for the study aim, (5) unbiased assessment of the study endpoint, (6) follow-up period appropriate for the study aim, (7) <5% loss to follow-up, and (8) prospective calculation of the sample size. For comparative studies, additional criteria were available: (9) adequate control group, (10) contemporary groups, (11) baseline equivalence of the groups, and (12) adequate statistical analyses. Discrepancies between the two investigators were resolved through discussion until a consensus was reached.

## 2.6. Statistical analysis

For continuous variables, the changing value between pre- and post-treatment was used for meta-analysis, and estimate was expressed using mean difference (MD) and 95% confidence interval (CI). For dichotomous variable, the number of event and total sample size were used for meta-analysis, and estimate was expressed using rate and 95% CI. Statistical heterogeneity between the included studies was evaluated using Cochrane $Q$ and Higgins' $I^2$. Statistical heterogeneity was deemed as significant if $I^2 > 50\%$ and $P < 0.1$ [42]. Nevertheless, the random-effects model was used for meta-analysis because it is not rational to ignore the variations between studies in the real settings [43, 44]. Subgroup analyses were conducted according to the subscales of tools. The possible publication bias was not assessed using funnel plots and Egger's test because the numbers of included studies were <10 in all analyses, in which case the funnel plots and Egger's test could yield misleading results [45]. Two-sided P-values <0.05 were considered statistically significant. All statistical analyses were conducted using STATA 14.0.

## 3. Results

### 3.1. Study selection

S1 Fig presents the study retrieval and selection process. The initial search identified 207 records, but 102 duplicates and 27 records marked as ineligible by automation tools were excluded before the screening. Then, 78 records were screened, and 59 records were excluded. Nineteen reports were sought for retrieval for eligibility evaluation. Five studies were excluded due to ineligible intervention (n = 1), duplicate studies (n = 2), and ineligible outcomes (n = 2). Finally, fourteen studies [27, 46–58] were included for meta-analysis to investigate the therapeutic efficacy and safety of autologous MFAT for the treatment of knee OA. And five studies [59–63] were included for meta-analysis to explore the similarity and correlation between the comorbidity of OA and AS.

### 3.2. Basic characteristics and quality of the included studies

There were eleven prospective studies [27, 46, 48, 49, 52–58] and eight retrospective studies [47, 50, 51, 59–63]. Other basic characteristics of the fourteen included studies to investigate the therapeutic efficacy and safety of autologous MFAT for the treatment of knee OA can be found in Table 1. And the basic characteristics of the five included studies to explore the similarity and correlation between the comorbidity of OA and AS can be found in Table 2.

**Table 1. Basic characteristics of the 14 included studies.**

| Study | Country | Study design | Mean age, years, | BMI, kg/m$^2$ | male % | Sample size | KOA degree | Donor site for adipose tissue | Device | Procedure | Follow-up, months |
|---|---|---|---|---|---|---|---|---|---|---|---|
| Boric, 2019 [50] | Croatia | Prospective | 69±12 | NR | 70.00 | 17 | NR | Abdominal subcutaneous | Lipogems® kit | Intra-articular, 4–15 mL | 24 |
| Cattaneo, 2018 [48] | Italy | Retrospective | 53.8 | 27±4 | 60.00 | 35 | NR | Lower or the lateral abdomen | Lipogems® kit | Intra-articular, 10 cm$^3$ | 12 |
| Genechten, 2021 [54] | Belgium | Prospective | 54.2 | 27.2 ±4.5 | 48.40 | 64 | NR | Lumbar region of the abdomen | Lipogem® kit | Injected under ultra-sound guidance from the lateral side of the index knee with an 18-gauge needle | 12 |
| Heidari, 2020 [52] | Italy | Prospective | <50 | NR | 54.50 | 110 | KL grade of III or IV, 80% | Lower abdominal area | Lipogems® kit | Single 6–8 ml injected directly into the knee joint under ultrasound guidance | NR |
| Hudetz, 2019 [27] | Croatia | Prospective | NR | <30, 13(65) | 75.00 | 20 | III and IV | Abdominal subcutaneous | Lipogems® kit | Intra-articular, 5 mL | 12 |
| Malanga, 2020 | USA | Prospective | 59.8 | 28.6 ±4.8 | 55.00 | 20 | mild to moderate | Abdomen | Lipogems® kit | Direct ultrasound guidance into the hypoechoic defects using primarily an 18-gauge 3-inch needle attached to a 3-mL syringe. | 12 |
| Mautner, 2019 [51] | USA | Retrospective | 63 | NR | 34.29 | 35 | NR | Abdomen | Lipogems® kit | 9 cc of MFAT was injected into the knee joint | 13.08 |
| Russo, 2017 [47] | Italy | Retrospective | 43 | 26 (24–28) | 70.00 | 30 | NR | Lower or lateral abdomen | Lipogems® kit | Intra-articular, 10–15 cm$^3$ | 12 |
| Baria, 2022 [55] | USA | Prospective | 56.1 | 31.0 | 28.60 | 28 | I to IV | Subcutaneous tissue of the abdomen or flank | Lipogems® kit | Intra-articular, 7.92 mL | 6 |
| Gobbi, 2022 [56] | Italy | Prospective | 62.75 | NR | 42.50 | 40 | I and II | Abdomen | Lipogems® kit | Intra-articular | 24 |
| Screpis, 2022 [57] | Italy | Prospective | 54.0 | 26.8 | 48.02 | 202 | I to IV | Lower abdomen or flank areas | Lipogems® kit | Intra-articular | 24 |
| Zaffagnin, 2022 [58] | Italy | Prospective | 54.5 | 25.9 | 52.83 | 53 | I to IV | Lower or lateral abdomen | Lipogems® kit | Intra-articular, 5 mL | 24 |
| Liang, 2018 [49] | China | Prospective | 52.31 | NR | 43.33 | 30 | NR | NR | Lipogems® kit | Intra-articular, 5 mL | NR |
| Li, 2017 [46] | China | Prospective | 53.81 | 29.20 | 37.04 | 27 | NR | Abdomen | Lipogems® kit | Intra-articular, 5 mL | 6 |

MFAT: micro-fragmented adipose tissue; NR: not reported; KOA: knee osteoarthritis.

Based on the MINORS tool, six studies scored 10 points [47, 51, 57, 59–63], three studies scored 11 points [48, 49, 52], six studies scored 12 points [27, 53–55], two studies scored 13 points [46, 50], and two studies scored 15 points [56, 58]. The detailed information regarding the level of evidence is summarized in S2 Table.

### 3.3. Knee injury and osteoarthritis outcome score

The mean absolute differences in KOOS subscale scores between pre- and post-treatment were 19.65 (13.66, 25.63), 14.20 (4.64, 23.76), 19.95 (13.02, 26.89), 25.23 (14.80, 35.66), and 26.01 (13.68, 38.35) for pain, symptoms, ADL, sports/recreation, and QOL, respectively. All pooled results are depicted in Fig 1. In addition, the results of individual studies which were included in the meta-analysis of KOOS subscale scores are displayed in S2 Fig.

**Table 2. Basic characteristics of the 5 included studies.**

| Author, year of publication | Study design | Sample size (OA) | Joint | OA evaluation | AS evaluation | Association |
|---|---|---|---|---|---|---|
| Jonsson, 2011 [59] | Cross sectional | 5170 | Knee, hip, | Joint replacement of knee and hip documented by CT scans, HOASCORE of hand photography | Intima media thickness of carotid by means of US, Agatston score of coronary and aorta calcification by means of CT angiography | NS for hip OA (OR 1.02 [0.64–1.62] for males, OR 1.07 [0.75–1.50] for females) NS for knee OA (OR 0.92 [0.55–1.51] for males, OR 1.49 [0.98–2.27] for females) |
| Hoeven, 2013 [60] | Cohort | 5650 | Knee, hip, hand | Kellgren-Lawrence grading of knee, hip, and hand x-ray | Intima media thickness of carotid by means of US | females only (OR 1.4 [1.19–1.65] for DIP OA, OR 1.5 [1.09–2.18] for MCP OA) NS for hip OA Positive for knee OA in females only (OR 1.7 [1.10–2.73]) |
| Hoeven, 2015 [61] | Cohort | 3465 | Knee | Kellgren-Lawrence grading of knee x-ray | Agatston score of aorta calcification by means of CT angiography | NS (OR 1.03 [0.92–1.15]) |
| Gielis, 2017 [62] | Cohort | 763 | Knee, hip | Kellgren-Lawrence grading of knee and hip x-ray | Semi-quantitative calcification score of iliac, femoral, popliteal and crural arteries by means of x-ray | Positive for females (OR 2.51 [1.57–4.03]) NS for males (OR 0.83 [0.41–1.68]) |
| Hussain, 2015 [63] | Cohort | 2476 | Knee | Coding for total knee replacement | Retinal vessel calibre by means of fundus photography | Positive (OR 2.02 [1.03–3.94]) |

OA–osteoarthritis, AS–atherosclerosis, NS–not significant, CT–computed tomography, OR–odds ratio, NA—not available, US–ultrasound, MRI–magnetic resonance imaging, DIP–distal interphalangeal joint, MCP–metacarpophalangeal joint

### 3.4. Pain

The mean differences in pain scores between pre- and post-treatment were -8.24 (-10.66, -5.82), -3.61 (-4.49, -2.72), -4.17 (-4.89, -3.44), and -2.17 (-4.06, -0.28) for VAS, resting VAS, activity VAS, and NRS, respectively. The pooled results are depicted in Fig 2.

### 3.5. Quality of life

The mean difference in QoL between pre- and post-treatment was -24.81 (-40.80, -8.82), 0.07 (0.02, 0.12), and 0.30 (-0.42, 1.02) for the Western Ontario and McMaster Universities Osteoarthritis Index (WOMAC), EQ-5D, and University of California in Los Angeles (UCLA), respectively. The pooled results are depicted in S3 Fig.

### 3.6. Adverse events

The use of MFAT was associated with the risk of bruising (12%), bleeding (5%), hematoma (15%), drainage (5%), infection (5%), soreness (50%), swelling (38%), pain (47%), and stiffness (55%) in harvest and injection sites. The pooled results are depicted in S4 Fig.

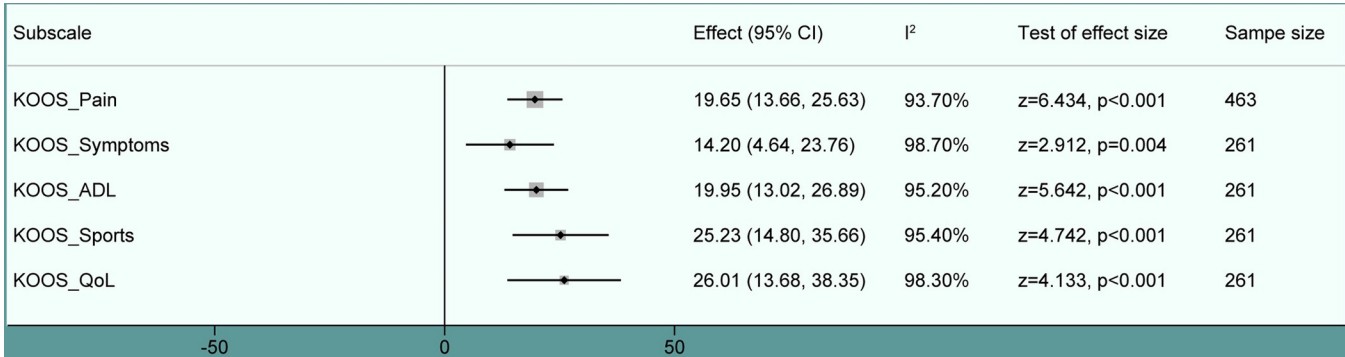

**Fig 1. Forest plot of subscales scores of the Knee injury and Osteoarthritis Outcome Score (KOOS).** ADL, activities of daily living; QoL, quality of life; CI, confidence interval.

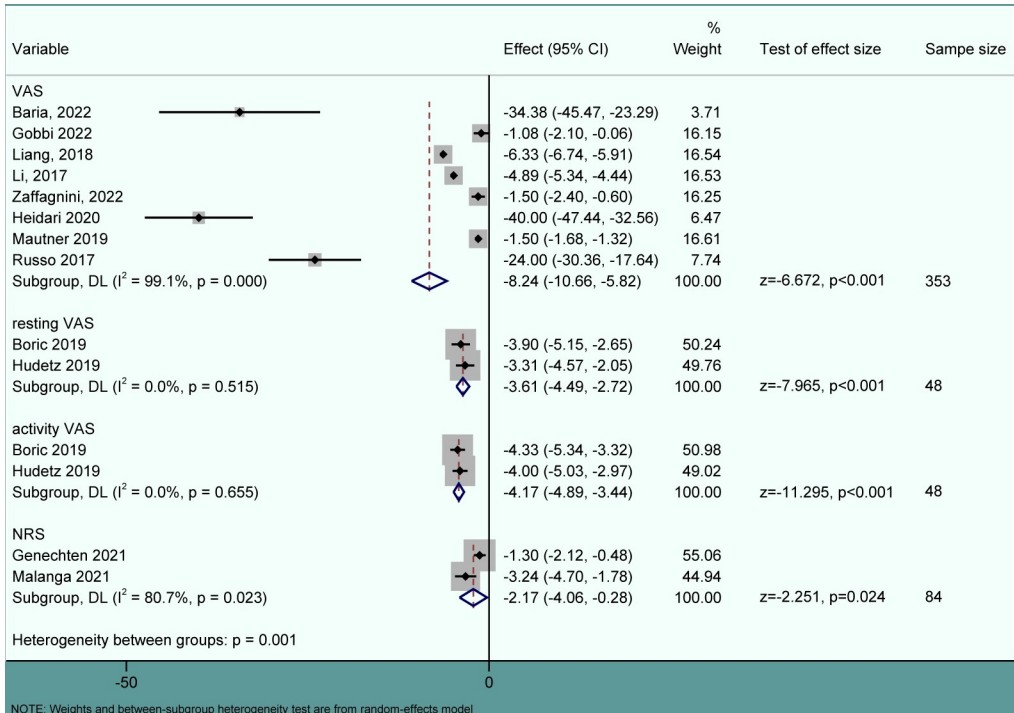

**Fig 2. Forest plot of pain scores.** VAS, visual analog scale; NRS, numerical rating scale; CI, confidence interval; DL, Dersimonian-Larid.

### 3.7. Tegner and IKDC

The mean difference in Tegner scoring system and International Knee Documentation Committee (IKDC) between pre- and post-treatment was 0.67 (-062, 1.97) and 13.70 (6.35, 21.04), respectively. The pooled results are depicted in S5 Fig.

### 3.8. Outcomes that could not be pooled

S6 Fig presents those outcomes that could not be pooled for meta-analysis. As shown in S6 Fig, the use of MFAT could be significantly associated with changes in OKS and HSS but not EQoL and dGEMRIC index.

### 3.9. Osteoarthritis and atherosclerosis

The metaanalysis evidenced a positive association between AT and OA (OR 1.17, CI 1.01–1.36). The pooled results are depicted in Fig 3.

## 4. Discussion

Both myocardial infarction and ischemic stroke are most often caused by atherosclerosis. To date, three systematic reviews have been published on the increased risk of OA and CVD [64–66]. The first systematic review by Hall et al. [64] examined an increased risk of cardiovascular disease given the presence of OA, based on searches prior to November 2014. They included 15 studies. The meta-analysis showed that patients with OA were more likely to develop heart failure (relative risk [RR] 2.8; 95% confidence interval [CI] 2.25–3.49), ischemic heart disease (RR: 1.78; 95% CI 1.18 to 2.69), or CVD (RR: 1.69; 95% CI 1.13 to 2.53), but the risk of stroke was not high (RR: 1; 95% CI 0.13–7.87), and the results of the latter showed great

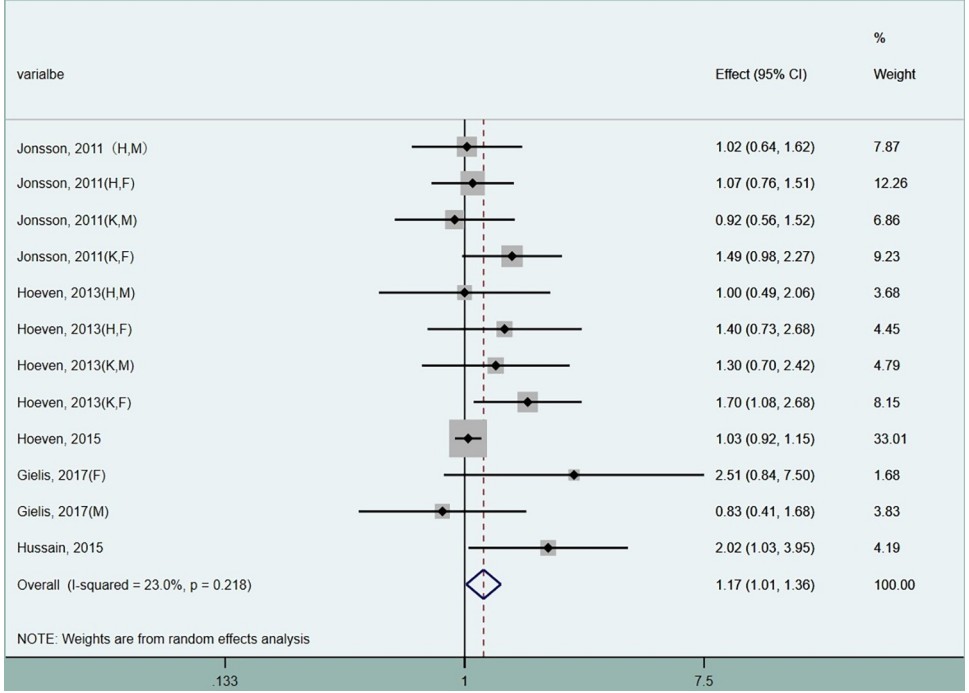

**Fig 3. Forest plot of the association between atherosclerosis and osteoarthritis.** OA, osteoarthritis; OR, odds ratio; CI, confidence interval; H, hips; K, knees; F, female; M, male.

heterogeneity. All studies included in the meta-analysis were longitudinal studies and measured confounding factors were taken into account. The second systematic review by Wang et al. [65], retrieved May 2016, also included 15 studies. In this review, cardiovascular disease risk from 10 prospective studies and 5 retrospective studies are presented. Overall, there was an increased risk of CVD for OA in prospective studies (RR 1.24; 95 CI 1.12 to 1.37), but not in retrospective studies (RR 1.15; 95% CI 0.95 ~ 1.38). Veronese et al. [66] reported the risk of cardiovascular death due to the presence of any joint osteoarthritis from studies published prior to November 2015, with a hazard ratio (HR) of 1.21 (95% CI 1.10 to 1.34) and a high degree of homogeneity in results. In short, most studies show cross-sectional and longitudinal associations, although not for all joints and all measures of atherosclerosis. Evidence from most studies suggests that atherosclerosis is associated with hand and knee osteoarthritis, but never hip osteoarthritis. Moreover, in studies analyzing gender differences, these associations were stronger in women than in men.

MFAT combined with surgery may be a promising therapeutic strategy for the early management of knee OA, but strong evidence for supporting the clinical application of MFAT is lacking. Therefore, the present meta-analysis was conducted to comprehensively investigate the therapeutic efficacy and safety of autologous MFAT for the treatment of knee OA. The pooled results reveal that MFAT may benefit to improve symptoms of knee OA although may also lead to some mild adverse events.

The present meta-analysis showed that post-operative application of MFAT significantly improve the five subscale -scores of the KOOS, pain scores, QoL scores, and Tegner score. It is consistent with the fact that all included studies reported some degree of improvements after the use of MFAT in their patients with knee OA [27, 46–50, 51–58]. Still, because most included studies have positive results, a publication bias is possible but could not be examined

in the present study because of the insufficient number of included studies for individual analysis [45]. Nevertheless, other studies that were not eligible for the present meta-analysis support the results of the present meta-analysis, suggesting that the use of MFAT may be promising for the treatment of knee OA [67, 68]. MFAT also showed benefits in other indications, such as menopausal vaginal atrophy, perianal fistula repair, and diabetic foot [48, 69–71].

Conventional treatments of knee OA is corticosteroid injection, but an inflammatory flare occurs in 2%-25% of the cases [7]. In addition, hyaluronic acid injection carries a risk of flares and granulomatous inflammation [72]. In the present meta-analysis, the meta-analysis of AEs reveals that the use of MFAT may lead to the occurrence of bruising, bleeding, hematoma, drainage, infection, soreness, pain, and stiffness; however, all adverse events were mild and relieved in the short term without specific treatment. In addition, whether the safety profile of MFAT is better than with other therapies (e.g., platelet-rich plasma) remains to be investigated in comparative trials.

Five studies [49–52, 58] included outcomes that could not be pooled in the present meta-analysis. These studies showed improvements in Oxford Knee Score (OKS) and Hospital for Special Surgery Knee Score (HSS), but not in the Emory Quality of Life (EQoL) score and delayed gadolinium-enhanced magnetic resonance imaging of cartilage (dGEMRIC) index. Of course, different assessment tools have different degrees of subjectivity/objectivity and measure different outcomes. Therefore, it might highlight the need to use multiple assessment tools and to use similar tools among studies to improve the comparability of the results.

This meta-analysis has limitations. The number of included studies was relatively small because the autologous MFAT strategy is relatively novel. Therefore, our findings should be interpreted with caution due to limited statistical power. All included studies used the same technique for harvest; however, there are rules or at least recommendations for the conduction of clinical and/or surgical trials in OA, which were not followed by the studies included in this systematic revie and meta-analysis. Indeed, the MINORS scores of the included studies were moderate. Significant heterogeneity was observed in several analyses, which would inevitably compromise the robustness and reliability of the pooled results. However, such heterogeneity was inevitable considering the differences among trials regarding severity of knee OA, donor sites for adipose tissue, techniques of intra-articular injection, and follow-up duration. We cannot perform subgroup analysis to explore which factors may contribute to significant heterogeneity due to limited number of eligible studies. Therefore, the results must be interpreted with caution. At the same time, most studies included in this systematic review and meta-analysis were retrospective studies, which had lower statistical power for assessing efficacy of a certain. Therefore, more prospective comparative or even randomized controlled trials with high-quality and large sample size are needed. Furthermore, these eligible studies did not assess changes in joint range of motion, a finding that is important for knee OA. Therefore, future studies should consider this result. Finally, there are many therapeutic options for the treatment of knee OA, such as sodium hyaluronate injection. However, it was impossible to quantitatively compared the efficacy and safety of intra-articular MFAT versus other therapeutic options due to lack of eligible studies. So, we recommend future studies to directly compared intra-articular MFAT to other therapeutic options.

## 5. Conclusions

In conclusion, based on the currently available limited evidence, there is a positive association between atherosclerosis and osteoarthritis; and the use of micro-fragmented adipose tissue may benefit for improving symptoms of knee osteoarthritis accompanied by atherosclerosis although may lead to some mild adverse events. However, more studies are necessary for

further validating our findings because some limitations such as limited number of eligible studies, significant statistical heterogeneity, and potential publication bias may significantly compromise the reliability of the pooled results.

## Supporting information

**S1 Checklist. PRISMA 2020 checklist.**
(DOCX)

**S1 Fig. Flow diagram of searching and selecting studies.** CNKI, China National Knowledge Infrastructure.
(TIF)

**S2 Fig. Subgroup analysis of KOOS subscale scores.** KOOS, Knee Injury and Osteoarthritis Outcome Score; ADL, activities of daily living; QoL, quality of life; CI, confidence interval.
(TIF)

**S3 Fig. Forest plot of quality of life (QOL).** WOMAC, the Western Ontario and McMaster Universities Osteoarthritis Index; EQ-5D, European Quality of Life Five Dimension Five Level; UCLA, University of California in Los Angeles (UCLA); CI, confidence interval; DL, Dersimonian-Larid.
(TIF)

**S4 Fig. Forest plot of adverse events (AEs).** ES, effect size; CI, confidence interval.
(TIF)

**S5 Fig. Forest plot of Tegner scoring system and IKDC. IKDC, International Knee Documentation Committee; CI, confidence interval; DL, Dersimonian-Larid.**
(TIF)

**S6 Fig. Forest plot of the outcomes that could not be meta-analyzed.** OKS, Oxford Knee Score; HSS, Hospital for Special Surgery Knee Score; EQoL, Emory Quality of Life; dGEMRIC, delayed gadolinium-enhanced magnetic resonance imaging of cartilage; CI, confidence interval; DL, Dersimonian-Larid.
(TIF)

**S1 Table. Quality assessment of the included studies.**
(DOCX)

**S2 Table. Quality assessment of the included studies.**
(DOCX)

## Author Contributions

**Conceptualization:** Wei Li.

**Data curation:** Wei Li.

**Formal analysis:** Wei Li.

**Funding acquisition:** Wei Li, Huajuan Guo.

**Investigation:** Wei Li, Huajuan Guo.

**Methodology:** Wei Li, Huajuan Guo.

**Project administration:** Wei Li, Congcong Wang.

**Resources:** Huajuan Guo, Congcong Wang, Yimin Zhang.

**Software:** Huajuan Guo, Congcong Wang, Yimin Zhang, Jun Wang.

**Supervision:** Congcong Wang, Yimin Zhang, Jun Wang.

**Validation:** Congcong Wang, Yimin Zhang, Jun Wang.

**Visualization:** Yimin Zhang, Jun Wang.

**Writing – original draft:** Jun Wang.

**Writing – review & editing:** Jun Wang.

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
