## [Decision Letter · Decision Letter 0]

9 Jun 2023

PONE-D-23-10633Autologous micro-fragmented adipose tissue in the treatment of atherosclerosis patients with knee osteoarthritis in Geriatric PopulationPLOS ONE

Dear Dr. Wang,

Thank you for submitting your manuscript to PLOS ONE. After careful consideration, we feel that it has merit but does not fully meet PLOS ONE’s publication criteria as it currently stands. Therefore, we invite you to submit a revised version of the manuscript that addresses the points raised during the review process.

We look forward to receiving your revised manuscript.

Kind regards,

Victor Afamefuna Egwuonwu, PhD

Academic Editor

PLOS ONE

Journal Requirements:

- https://www.sciencedirect.com/science/article/abs/pii/S0531556522000420?via%3Dihub

- https://www.sciencedirect.com/science/article/abs/pii/S1521694218300433?via%3Dihub

In your revision ensure you cite all your sources (including your own works), and quote or rephrase any duplicated text outside the methods section. Further consideration is dependent on these concerns being addressed.

3. Please identify your study as "systematic review and meta-analysis" in the title of your manuscript.

4. Please note that funding information should not appear in any section or other areas of your manuscript. We will only publish funding information present in the Funding Statement section of the online submission form. Please remove any funding-related text from the manuscript.

  "This work was supported by the National Natural Science Foundation of China (Grant number: No.82071470), the Natural Science Foundation of Shandong Province (Grant number: ZR2019PH026 and ZR2020MH094), and the Science and technology development plan of Weifang (Grant number: 2020YX014)."

Additional Editor Comments:

Dear Authors,

Please carefully attend to the reviewers concerns and comments regarding your submission and submit your corrections, rebuttal or comments. As quickly as you can to hasten the peer-review process.

Reviewers' comments:

Reviewer's Responses to Questions

**Comments to the Author**

1. Is the manuscript technically sound, and do the data support the conclusions?

Reviewer #1: Yes

2. Has the statistical analysis been performed appropriately and rigorously? 

Reviewer #1: Yes

3. Have the authors made all data underlying the findings in their manuscript fully available?

Reviewer #1: Yes

4. Is the manuscript presented in an intelligible fashion and written in standard English?

Reviewer #1: Yes

5. Review Comments to the Author

Reviewer #1: The study is interest. It is a meta-analysis study of Autologous micro-fragmented adipose tissue in the treatment of atherosclerosis patients with knee osteoarthritis in the elderly population. It needs Major Revision by a native English speaker.

6. PLOS authors have the option to publish the peer review history of their article (what does this mean?). If published, this will include your full peer review and any attached files.

Reviewer #1: No

---

## [Editor Report · Decision Letter 1]

24 Jul 2023

Autologous micro-fragmented adipose tissue in the treatment of atherosclerosis patients with knee osteoarthritis in Geriatric Population: a systematic review and meta-analysis

PONE-D-23-10633R1

Dear Dr. Wang,

We’re pleased to inform you that your manuscript has been judged scientifically suitable for publication and will be formally accepted for publication once it meets all outstanding technical requirements.

Kind regards,

Victor Afamefuna Egwuonwu, PhD

Academic Editor

PLOS ONE

Additional Editor Comments (optional):

Dear Wang,

I am writing to confirm my acceptance of your manuscript for publication.

Thank you for your decision to publish with plos one.
---

## [Editor Report · Acceptance letter]

22 Aug 2023

PONE-D-23-10633R1 

Autologous micro-fragmented adipose tissue in the treatment of atherosclerosis patients with knee osteoarthritis in Geriatric Population: a systematic review and meta-analysis 

Dear Dr. Wang:

I'm pleased to inform you that your manuscript has been deemed suitable for publication in PLOS ONE. Congratulations! Your manuscript is now with our production department. 

Kind regards, 

on behalf of

Dr. Victor Afamefuna Egwuonwu 

Academic Editor

PLOS ONE